# Multimeric single-domain antibody complexes protect against bunyavirus infections

**Paul J Wichgers Schreur[1]\*, Sandra van de Water[1], Michiel Harmsen[1], Erick Bermúdez-Méndez[1,2], Dubravka Drabek[3,4], Frank Grosveld[3,4], Kerstin Wernike[5], Martin Beer[5], Andrea Aebischer[5], Olalekan Daramola[6], Sara Rodriguez Conde[6], Karen Brennan[6], Dorota Kozub[6], Maiken Søndergaard Kristiansen[6], Kieran K Mistry[6], Ziyan Deng[6], Jan Hellert[7], Pablo Guardado-Calvo[7], Félix A Rey[7], Lucien van Keulen[1], Jeroen Kortekaas[1,2]**

[1]Department of Virology, Wageningen Bioveterinary Research, Lelystad, Netherlands; [2]Laboratory of Virology, Wageningen University, Wageningen, Netherlands; [3]Department of Cell Biology, Erasmus MC, Rotterdam, Netherlands; [4]Harbour Antibodies B.V, Rotterdam, Netherlands; [5]Institute of Diagnostic Virology, Friedrich-Loeffler-Institut, Greifswald - Insel Riems, Germany; [6]Biopharmaceutical Development, R&D BioPharmaceuticals, AstraZeneca, Cambridge, United Kingdom; [7]Structural Virology Unit, Virology Department, CNRS UMR 3569, Institut Pasteur, Paris, France

**\*For correspondence:**
paul.wichgersschreur@wur.nl

**Abstract** The World Health Organization has included three bunyaviruses posing an increasing threat to human health on the Blueprint list of viruses likely to cause major epidemics and for which no, or insufficient countermeasures exist. Here, we describe a broadly applicable strategy, based on llama-derived single-domain antibodies (VHHs), for the development of bunyavirus biotherapeutics. The method was validated using the zoonotic Rift Valley fever virus (RVFV) and Schmallenberg virus (SBV), an emerging pathogen of ruminants, as model pathogens. VHH building blocks were assembled into highly potent neutralizing complexes using bacterial superglue technology. The multimeric complexes were shown to reduce and prevent virus-induced morbidity and mortality in mice upon prophylactic administration. Bispecific molecules engineered to present two different VHHs fused to an Fc domain were further shown to be effective upon therapeutic administration. The presented VHH-based technology holds great promise for the development of bunyavirus antiviral therapies.

## Introduction

Facilitated by globalization and climate change, arthropod-borne viruses (arboviruses) increasingly pose a threat to human and animal health (*Gould et al., 2017*). Although for some arboviruses vaccines are available that can be used to prevent or control outbreaks, for the vast majority of emerging arboviruses no countermeasures are available. Rift Valley fever virus (RVFV), a phlebovirus within the order *Bunyavirales*, is prioritized by the World Health Organization (WHO) as being likely to cause major epidemics for which no, or insufficient countermeasures exist. RVFV is currently confined to the African continent, the Arabian Peninsula and several islands off the coast of Southern Africa, where it causes recurrent outbreaks (*Clark et al., 2018*). The world-wide distribution of competent mosquito vectors and susceptible animals underscores the risk for emergence in currently unaffected areas. In endemic areas, RVFV causes major epizootics among livestock, characterized by abortion

storms and large-scale mortality among newborn ruminants. Importantly, the virus also infects humans, either through direct contact with infected animal tissues or via the bites of infected mosquitoes (*Hartman, 2017*). Infected individuals generally present with mild to severe flu-like symptoms, however a minority of patients may develop encephalitis or hemorrhagic fever.

In addition to members of the family *Phenuiviridae*, like RVFV, members of the family *Peribunyaviridae*, genus *Orthobunyavirus*, may also cause severe disease in humans and animals. In 2011, the incursion of Schmallenberg virus (SBV) in Europe demonstrated that orthobunyaviruses are capable of spreading very efficiently across new territories. SBV infections are associated with fever and reduced milk production in cows and severe malformations in offspring of both large and small ruminants (*Hoffmann et al., 2012*; *van den Brom et al., 2012*). SBV is categorized as a biosafety level-2 pathogen and is not pathogenic to humans, facilitating its use as a model of zoonotic orthobunyaviruses that require a higher level of containment (*Golender et al., 2015*; *van Eeden et al., 2014*; *van Eeden et al., 2012*).

At the moment, no effective antiviral therapy is available to treat bunyavirus infections in humans. Patients essentially rely on supportive care. Some promising pre-clinical and clinical data have been obtained with small molecules, mainly nucleoside analogues, that interfere with viral replication (*Gowen and Hickerson, 2017*). Recent X-ray crystallography and cryo-electron microscopy/tomography data have provided novel insights into the bunyavirus glycoprotein architecture that may facilitate the development of bunyavirus antibody therapies in the future (*Wu et al., 2017b*; *Hellert et al., 2019*; *Halldorsson et al., 2018*). After vaccines, antibody therapies are considered the most effective tools to fight (re)emerging life-threatening viral infections (*Jin et al., 2017*).

A very wide range of approaches are currently used to isolate natural antibodies and to design synthetic constructs. Most of these efforts are based on neutralizing antibodies or antibody fragments of murine or human origin. In general, these molecules consist of heavy and light polypeptide chains linked by disulphide bonds. Interestingly, in only a few species such as camelids and nurse sharks, heavy chain-only antibodies (HCAbs) are found, of which the antigen binding domain can be expressed as a single-domain antibody (sdAb) (*Arbabi-Ghahroudi, 2017*). Camelid derived sdAbs, known as VHHs, are increasingly used as tools in medicine, including virus neutralization (*Wu et al., 2017a*; *De Vlieger et al., 2018*). VHHs are intrinsically highly soluble molecules and due to their distinctive structure with extended antigen-binding CDR3 region and overall small size, some are able to target unique (cryptic) antigenic sites not accessible to conventional antibodies (*Muyldermans, 2013*). VHHs with nanomolar or even picomolar affinity targeting a broad spectrum of antigens have been described (*Jin et al., 2017*; *Arbabi-Ghahroudi, 2017*). The single-domain nature additionally allows easy genetic manipulation and favors efficient expression in various heterologous systems including microorganisms (*Harmsen and De Haard, 2007a*; *Liu and Huang, 2018*).

So far, VHHs have been explored as tools for a wide range of applications including their use as therapeutic agents (*Wu et al., 2017a*; *Harmsen and De Haard, 2007a*; *Bannas et al., 2017*; *Traenkle and Rothbauer, 2017*; *Gonzalez-Sapienza et al., 2017*). With respect to virus neutralization, a multimeric VHH has shown great promise in counteracting severe pulmonary disease in infants caused by respiratory syncytial virus (RSV) (*Detalle et al., 2016*). The therapeutic potency of complexes consisting of multimers of the same VHH (multivalent) or combinations of VHHs targeting different antigenic sites (multispecific) is explained by their improved avidity, selectivity and kinetics compared to individual VHHs (*Hultberg et al., 2011*). Although techniques to use VHHs as building blocks to generate multifunctional molecules are well established, selection of the optimal VHH combination and optimal VHH formats is still challenging (*Iezzi et al., 2018*).

Recently, the discovery of 'bacterial superglues' has enabled the creation of unique protein architectures. The superglues are comprised of a bacterially-derived peptide and a small protein able to form unbreakable isopeptide bonds (*Veggiani et al., 2014*). The first bacterial superglue was developed by splitting the immunoglobulin-like collagen adhesion domain (CnaB2) of the fibronectin binding protein (FbaB) of *Streptococcus pyogenes* (*Zakeri et al., 2012*) into a peptide and a protein fragment referred to as SpyTag and SpyCatcher. When the two peptides meet, an amide bond is formed which is highly specific and irreversible. Covalent bond formation occurs within minutes upon mixing and is highly robust under various conditions (*Zakeri et al., 2012*). The potency of SpyTag:SpyCatcher has already found applications in various disciplines and triggered the search for various additional protein and peptide partners (*Veggiani et al., 2016*; *Tan et al., 2016*).

Here, we selected and characterized RVFV and SBV-specific VHHs targeting receptor binding glycoprotein domains, and used bacterial superglues to create virus neutralizing VHH complexes that reduced and prevented morbidity and mortality in mouse models. Selected RVFV VHHs were subsequently reformatted into human Fc-based bispecific chimeric HCAbs that showed protection in pre- and post-exposure treatments.

## Results

### Selection of RVFV and SBV-specific VHHs

To obtain a source of RVFV and SBV-specific VHHs with potential neutralizing activity for the respective viruses, llamas were inoculated with the live-attenuated RVFV vaccine strain Clone 13, RVFV virus-like particles (VLPs) or SBV NL-F6 (*Figure 1A*). All inoculated animals responded well with virus-specific neutralizing antibodies in serum above a 50% neutralizing titer of 500 (*Figure 1B*; *Wichgers Schreur et al., 2017*; *Loeffen et al., 2012*). The B-cell-derived VHH phagemid libraries contained $>10^7$ unique clones per animal.

RVFV and SBV-specific VHHs were isolated from VHH phage display libraries by pannings with the RVFV Gn ectodomain (RVFV-Gn$_{ecto}$) (*de Boer et al., 2010*) or the SBV Gc head domain (SBV-Gc$_{head}$) (*Figure 1C*; *Hellert et al., 2019*), previously described as the Gc *N*-terminal subdomain of 234 amino acids (Gc Amino) (*Wernike et al., 2017*). Both RVFV-Gn$_{ecto}$ and SBV-Gc$_{head}$ are considered to be the main targets of neutralizing antibodies against the respective viruses (*Hellert et al., 2019*; *Wernike et al., 2017*; *Roman-Sosa et al., 2017*; *Kortekaas et al., 2010*; *Kortekaas et al., 2012*). Supernatants of induced clones were screened for RVFV-Gn$_{ecto}$ and SBV-Gc$_{head}$ specificity by indirect ELISA and immunoperoxidase monolayer assay (IPMA). In total 62 unique RVFV-Gn$_{ecto}$-specific clones that grouped into 16 CDR groups (15 CDR3 groups) and 15 unique SBV-Gc$_{head}$-specific clones that grouped into 6 CDR3 groups were identified (*Figure 1D*; *Figure 1—figure supplement 1*). One representative VHH per CDR group was subsequently produced in yeast (*Figure 1E*). All yeast expressed VHHs were confirmed to recognize soluble RVFV-Gn$_{ecto}$, or SBV-Gc$_{head}$ by ELISA (*Figure 1F,G*) and/or were found to bind to fixed viral antigen in infected cells (*Figure 1—figure supplement 2*).

### RVFV and SBV VHHs recognize distinct antigenic sites

To determine the number of antigenic sites targeted by the different VHHs, RVFV-Gn$_{ecto}$ and SBV-Gc$_{head}$-specific competition ELISAs were performed. The RVFV VHH panel (N = 16) was shown to target three independent antigenic sites within RVFV-Gn$_{ecto}$, referred to as the red, blue and yellow antigenic sites (*Figure 2A*). When compared to the VHHs of the blue and the yellow antigenic sites, all of the VHHs of the red antigenic site did not show typical sigmoidal curves, suggesting markedly reduced avidity for monomeric Gn$_{ecto}$ (*Figure 1F*). The SBV-specific VHHs showed a more complex pattern of competition (*Figure 2B*). SB10 (beige) clearly targets an antigenic site that does not overlap with the sites targeted by the other five VHHs. SB12 and SB13 (orange) recognize the same antigenic site that does not overlap with the site targeted by SB11 (green). However, SB9 and SB14 (violet) show reciprocal competition with each other and SB9 competes (reciprocally) with SB11. SB14 shows non-reciprocal competition with SB12 and SB13, which strongly suggests that the epitopes of SB9, SB11, SB12, SB13 and SB14 are all in very close proximity (*Figure 2B*).

### Mapping of the RVFV VHHs to the RVFV-Gn head domain

The N-terminal 154–469 amino acids of RVFV-Gn$_{ecto}$, now known as the RVFV-Gn head domain, was recently crystallized and shown to comprise three subdomains; subdomain A (or I), β ribbon domain (II) and domain B (III) (*Wu et al., 2017b*; *Halldorsson et al., 2018*). To assess whether the VHHs target a specific subdomain, the head domain as well as the individual subdomains were expressed using the baculovirus expression system. The subdomains were expressed successfully (*Figure 2C*) and binding to the RVFV VHHs was assessed by indirect ELISA (*Figure 2D*). The results show that the yellow antigenic site, which is targeted by RV104, RV136 and RV148, is located on subdomain B and that the blue antigenic site, recognized by RV107 and RV115, is primarily located on the β-ribbon domain. The red antigenic site could not be mapped to a specific subdomain, suggesting that this site is present at a subdomain interface. Gn, as part of the Gn/Gc heterodimer, is organized

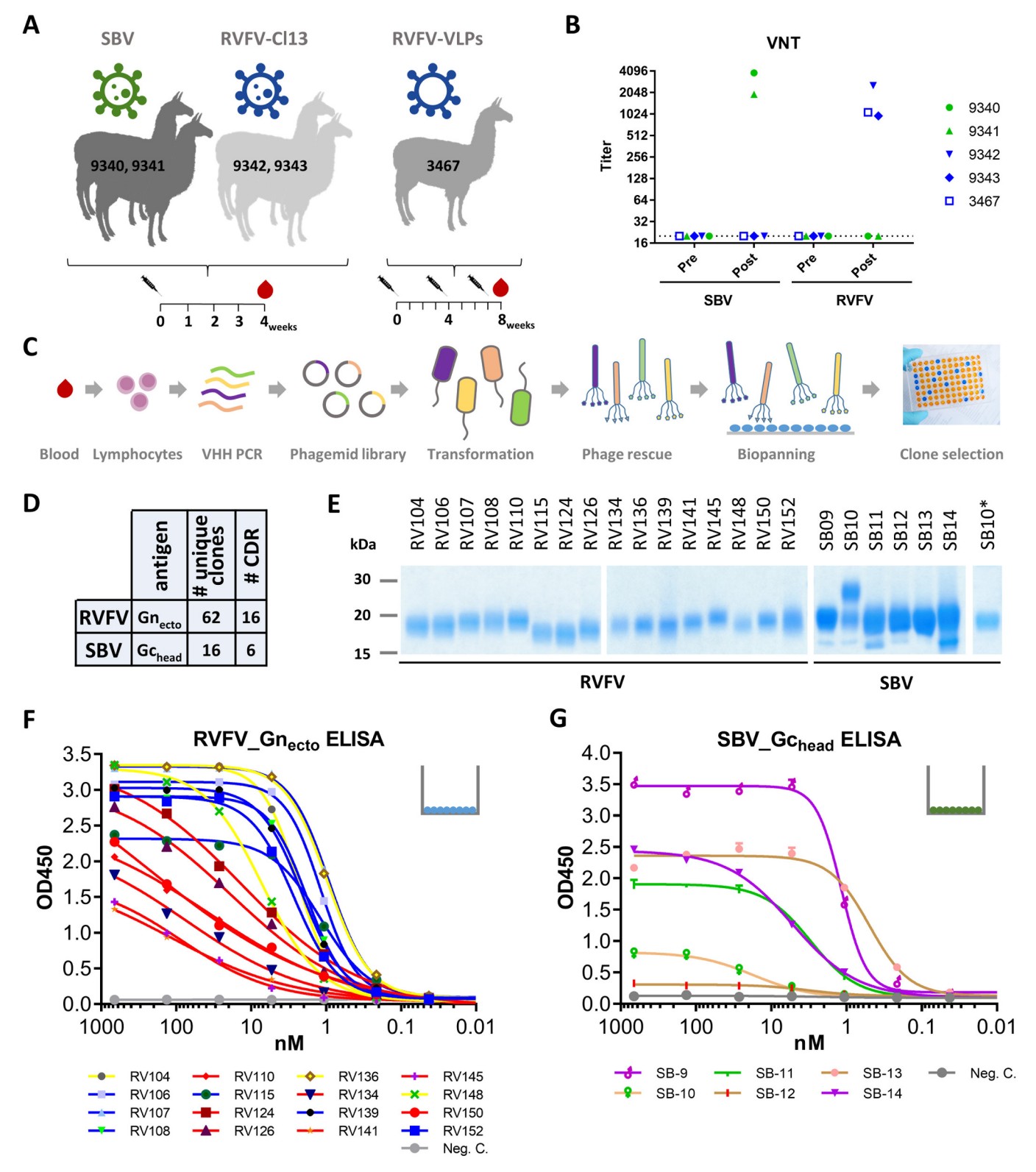

**Figure 1.** Selection and expression of RVFV and SBV-specific VHHs. (**A**) Illustration of the llama immunization strategy to obtain a source for RVFV and SBV-specific VHHs. (**B**) RVFV and SBV-specific virus neutralizing responses in llama sera determined by virus neutralization test (VNT) before (pre) and post immunization. Neutralizing antibody titers were measured by endpoint titration and calculated as 50% neutralization. (**C**) Schematic presentation of the VHH library construction and phage display selections. Blood lymphocyte RNA was used as a template for VHH-specific PCRs and purified

*Figure 1 continued on next page*

*Figure 1 continued*

fragments were cloned into phagemids generating a phagemid library. Following phagemid transduction, the purified phage library was screened for antigen-specific phages. (D) Summarizing table of the VHH clone selection showing the number (#) of unique clones identified and the number of CDR groups per antigen. (E) SDS-PAGE of yeast-expressed RVFV and SBV-specific VHHs. SB10 was shown to be partly N-glycosylated as confirmed by PNGase F based deglycosylation (lane SB10*). (F) RVFV-Gn$_{ecto}$ and (G) SBV-Gc$_{head}$-specific indirect ELISAs. The color coding of the individual VHHs is based on the outcome of the competition ELISA result as presented in *Figure 2A and B*.

The online version of this article includes the following figure supplement(s) for figure 1:

**Figure supplement 1.** VHH amino acid sequence alignment.

**Figure supplement 2.** VHH specificity determined by immunoperoxidase monolayer assay (IPMA).

similarly in both hexamers and pentamers on the RVFV particle, of which Gn is most exposed to the immune system. We mapped the antigenic sites to the previously proposed Gn structure within the pentameric capsomer density (*Figure 2F*).

## Mapping of the SBV VHHs on the SBV-Gc head domain

The SBV-Gc head domain is the most exposed feature on the virus envelope, and based on homology with other orthobunyaviruses, it is believed to reversibly form homotrimers among neighboring Gc molecules (*Hellert et al., 2019*). Crystal structures of the head domain in complex with two neutralizing murine mAbs, referred to as 1C11 and 4B6, have recently shown that 1C11 binds to the apical end of the head domain, near the N686 glycosylation site, and that 4B6 binds to the opposite, basal end of the head domain, near its connection to the stalk (*Hellert et al., 2019*). Binding of either antibody is sterically incompatible with head domain trimerization, suggesting that neutralization may at least be partially mediated by destabilization of the spike's quaternary structure (*Hellert et al., 2019*). Competition ELISAs with 1C11 and 4B6 and our SBV VHH panel suggests that SB11 and SB14 target the same antigenic site as 1C11 at the apical end of the domain, whereas no competition was observed with 4B6 (*Figure 2B*). For SB10, of which the antigenic site is independent from the others, we generated an escape mutant carrying the single point mutation Y541C located directly at the head domain's contact area with the stalk (*Figure 2E,G*). This site likely acts as a hinge allowing the head domain to adopt different orientations relative to the virion surface (*Hellert et al., 2019*), and antibody binding is thus expected to restrain the local mobility of the spike.

## Multimeric VHH complexes show superior neutralization activity

The virus neutralizing activity of selected VHHs, representative for each antigenic site, was evaluated with RVFV and SBV-specific virus neutralization tests (VNTs). At nM range concentrations, no neutralization was observed for any of the monovalent VHHs (*Figure 3*; left panel of the figure and diagonal set of bars). Since synergistic effects have been reported for combinations of monoclonal antibodies in bunyavirus neutralization (*Besselaar and Blackburn, 1992*), we subsequently tested mixtures of VHHs. Combining two VHHs in the VNT assays resulted in moderate neutralization for some SBV VHH combinations (ND$_{50}$ around 200 nM) whereas still no neutralization was observed for RVFV VHH mixtures (*Figure 3*; left part of the figure with double colored bars). We therefore assessed whether multimerization supports efficient neutralization. In order to facilitate the screening of a large set of bivalent and bispecific complexes including various linker lengths and without the need to construct a lot of expression plasmids, we used the advantageous properties of bacterial superglue. VHHs were expressed with Spy- and SnoopTags and flexible scaffolds consisting of one, two or three elastin-like protein (ELP) domains were expressed with Spy- and SnoopCatcher domains. By simply mixing the tagged VHH versions with the Catcher containing ELP scaffolds, bivalent and bispecific complexes were constructed (*Figure 4*). The VNT assays subsequently revealed that several multimeric complexes very efficiently neutralized with ND$_{50}$ <10 nM (*Figure 3*; middle and right panels). For RVFV, targeting of two distinct antigenic sites seemed a prerequisite to enable efficient neutralization (*Figure 3A*), whereas for SBV both bivalent and bispecific complexes were capable of neutralization (*Figure 3B*). Interestingly, the length of the scaffolds did not have a major effect on neutralization efficiency of the RVFV VHH complexes, whereas the length did influence efficacy of the SBV complexes. Complexes based on the ELP3 scaffold (the longest tested) showed markedly higher neutralization efficiencies.

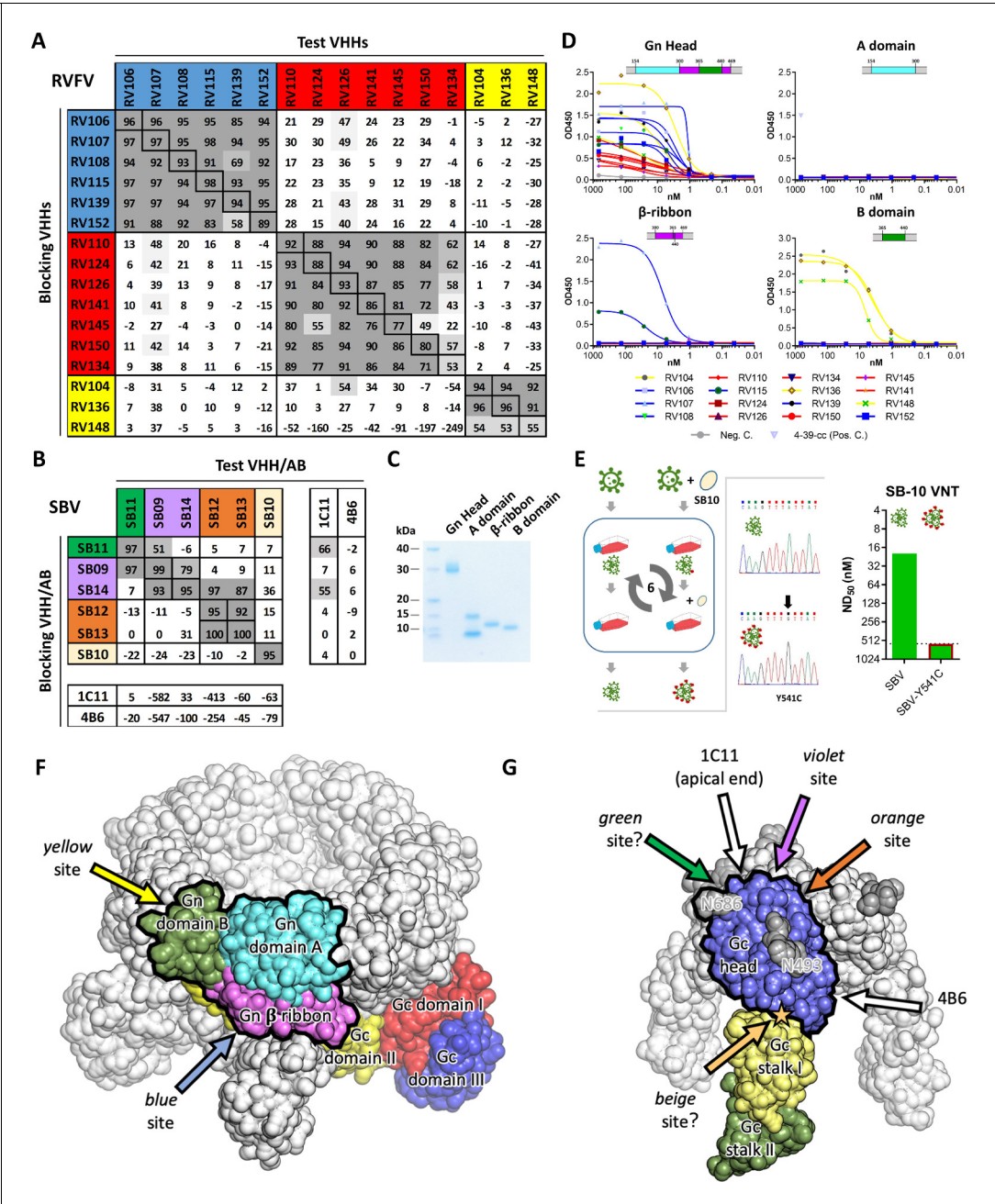

**Figure 2.** Characterization of VHH binding sites. (**A**) RVFV and (**B**) SBV-specific VHHs and antibodies were tested in competition ELISA using RVFV-Gn_ecto and SBV-Gc_head as antigens. Competition is expressed as percentage of blocking. A percentage above 50% is considered efficient blocking and strongly suggests that the test VHH binds at an overlapping site in the antigen as the blocking VHH. For SBV, competition was additionally assessed for the 1C11 and 4B6 mouse mAbs (*Hellert et al., 2019*). The RVFV-specific VHHs clustered into three groups each targeting a different antigenic site in RVFV-Gn_ecto and the SBV VHH panel was shown to target 4 different antigenic sites in SBV-Gc_head (of which the green, violet and orange sites are shown to be partly overlapping). Since SB11 and SB14 showed efficient competition (although non-reciprocal) with 1C11, the green, violet and orange antigenic sites are expected to be located at the apical end of the SBV-Gc_head domain. (**C**) SDS-PAGE of the baculovirus produced head, A, β-ribbon and B subdomains of RVFV-Gn_ecto. The lowest band in the subdomain A lane is expected to be the result of proteolytic cleavage. (**D**) RVFV-Gn_ecto subdomain-based VHH ELISAs. Wells were coated with the indicated subdomains and VHH binding was assessed with a HRP-conjugated goat anti-llama IgG. (**E**) SB10 escape mutant selection. Briefly, wildtype SBV was sequentially passaged in the absence or presence of SB10. Following passage 6, the Gc_head coding region was sequenced and SB10 neutralization was quantitatively assessed. (**F**) Model of a pentameric RVFV spike. The RVFV Gn/Gc heterodimer within a penton (PDB: 6F9F, *Halldorsson et al., 2018*) is color-coded by domains. The Gn head domain is outlined in black. Arrows indicate approximate positions of the two mapped antigenic sites. (**G**) Model of a trimeric SBV spike based on the monomeric SBV crystal structure (PDB: 6H3S, *Hellert et al., 2019*) and the trimeric BUNV cryo-ET map (EMDB: EMD-2352, *Bowden et al., 2013*). A protomer is color-coded by

*Figure 2 continued on next page*

*Figure 2 continued*

domains and the Gc head domain is outlined in black. The two N-glycans are labelled with the positions of their respective asparagine residues and the position of the SB10 escape mutation is indicated with a star. Arrows indicate approximate positions of the antigenic sites. Antigenic sites for which evidence is inconclusive are indicated with question marks (green and beige sites).

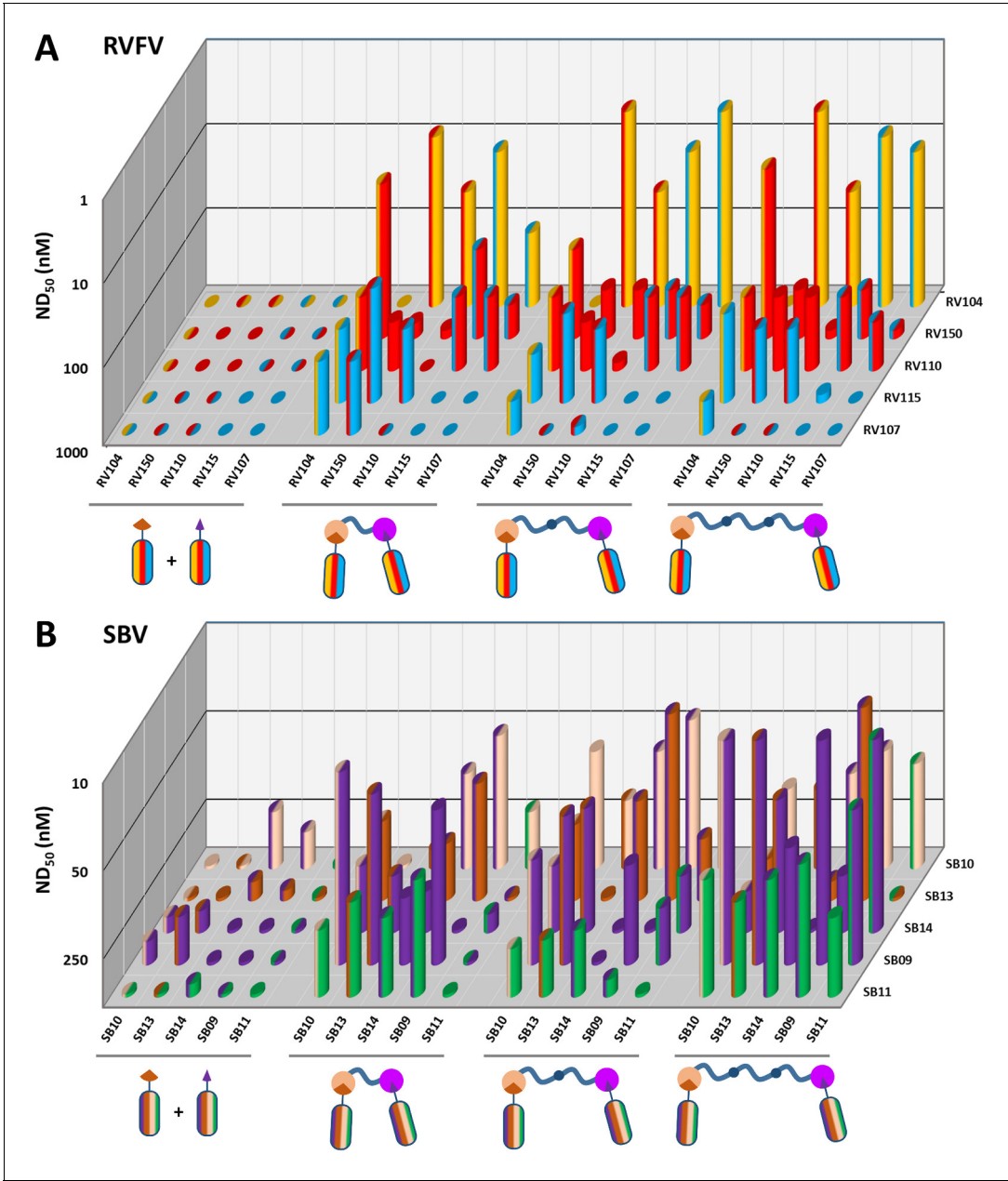

**Figure 3.** RVFV and SBV-specific neutralization by monovalent VHHs and bacterial superglue-assembled VHH complexes. (**A**) RVFV- and (**B**) SBV-specific monovalent VHHs, VHH combinations and bivalent and bispecific ELP-based VHH complexes were tested in a RVFV or SBV-specific VNT assay. Complexes were formed by mixing the tagged versions of the VHHs with the indicated Catcher-containing ELP scaffolds of different lengths (*Figure 4*). Mixtures were subsequently pre-incubated with virus before adding them to susceptible cells. The ND$_{50}$ values refer to total VHH per reaction as neutralization might be the result of cooperative action of VHH complexes and free VHHs. Averages of two biological replicate experiments are presented. Individual VHHs are color-coded based on the antigenic site they recognize as determined by competition ELISA (*Figure 2A and B*). Generally, most efficient neutralization was observed with multimerization of VHHs that each target a different antigenic site.

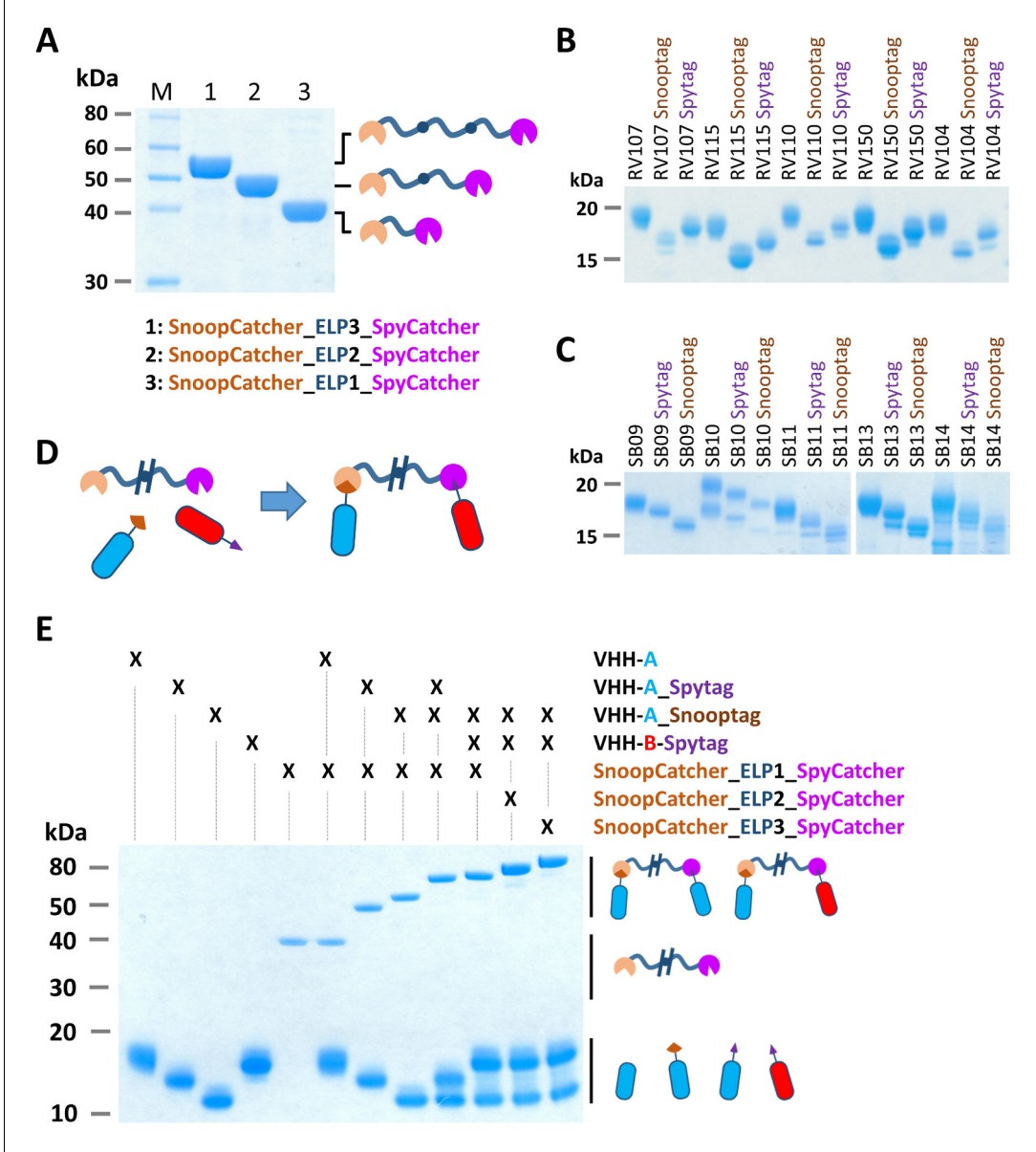

**Figure 4.** Bacterial superglue-based VHH formatting. (**A**) SDS-PAGE of ELP-based scaffolds with an N-terminal SnoopCatcher and a C-terminal SpyCatcher domain. (**B**) SDS-PAGE of RVFV and (**C**) SBV-specific VHHs expressed with a C-terminal Snooptag or Spytag. The untagged variants were expressed with the llama antibody long hinge region. (**D**) Illustration of the bacterial superglue-based site-directed formation of a bispecific or bivalent VHH complex. (**E**) SDS-PAGE of the bacterial superglue-based formation of bivalent and bispecific VHH complexes using ELPs of various length. VHH RV115 and VHH RV150 were used as an example for VHH-A and VHH-B, respectively.

## Prophylactic administration of ELP-based VHH complexes reduces morbidity and mortality in mice

Following the *in vitro* neutralization experiments, selected bivalent and bispecific complexes were evaluated in mouse infection experiments. To extend the *in vivo* half-life, the complexes were expressed with an albumin binding domain (ABD) (*Figure 5A,B,C*). Binding of a VHH complex to circulating albumin is expected to extend the half-life not only due to an increased size of the complex, but also as a result of recycling through the neonatal Fc receptor (FcRn) (*Mehand et al., 2018*; *Jacobs et al., 2015*). Additionally, one trispecific RVFV VHH complex was constructed and evaluated to take the bacterial superglue approach to the next level and to make a construct targeting the red, blue and yellow antigenic sites within the RVFV-Gn$_{head}$ domain simultaneously. To enable the

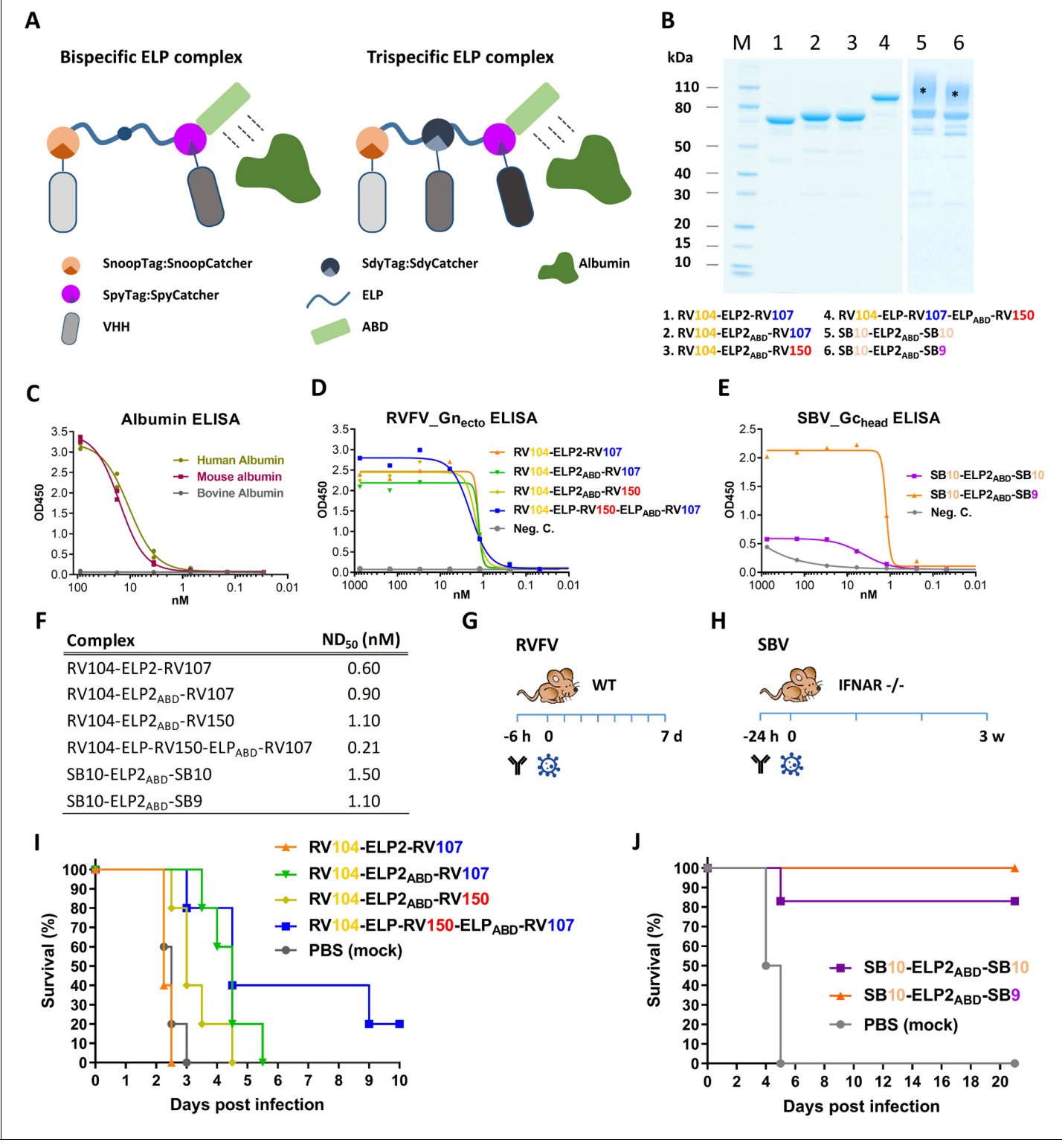

**Figure 5.** Efficacy of RVFV and SBV-specific ELP-based VHH complexes in mice. (**A**) Illustration of ELP-based VHH complexes comprising an albumin binding domain (ABD). (**B**) SDS-PAGE of purified RVFV and SBV-specific ELP-VHH complexes. Uncoupled VHHs were removed by a Strep-tag based purification following the coupling. The asterisks (*) indicate the glycosylated variants with higher molecular weights. (**C**) Species-specific albumin ELISA of the ABD containing ELP scaffold. Human, mouse or bovine albumin was coated at a 10 μg/ml concentration and the trispecific ELP scaffold containing an ABD domain (see **A**) was incubated in a dilution series. Binding was visualized with a Strep-Tactin-HRP conjugate. (**D**) Indirect RVFV-Gn$_{ecto}$ and (**E**) SBV-Gc$_{head}$ ELISA of purified RVFV and SBV-specific VHH complexes. (**F**) Neutralizing activity of the purified VHH complexes as

*Figure 5 continued on next page*

*Figure 5 continued*

determined by RVFV and SBV-specific VNTs. The $ND_{50}$ values refer to the concentration of complexes. (G) Illustration of the RVFV and (H) SBV mouse models. (I) Survival curve of RVFV infected mice pre-treated with 150 µg of the RVFV-specific VHH complexes. (J) Survival curve of SBV-infected mice pre-treated with either 200 µg of SB10-ELP2$_{ABD}$-SB10 or 100 µg SB10-ELP2$_{ABD}$-SB9.

The online version of this article includes the following figure supplement(s) for figure 5:

**Figure supplement 1.** ELP-based SBV-specific VHH complexes reduce viremia.

construction of this trispecific complex we made use of a third bacterial superglue; SdyTag and Sdy-Catcher (*Veggiani et al., 2014*). The specificity of all multimeric complexes to bind to RVFV-Gn$_{ecto}$ or SBV-Gc$_{head}$ was confirmed by indirect ELISA (*Figure 5D,E*) and efficient neutralization at nanomolar scale was confirmed by VNT (*Figure 5F*).

Mice were inoculated once with the complexes, followed by a lethal challenge dose of either RVFV or SBV. RVFV-specific complexes were evaluated in a BALB/c mouse model and SBV-specific complexes in IFNAR$^{-/-}$ mice (*Figure 5G,H*; *Wernike et al., 2012*). The results of the RVFV mouse experiment showed that in the absence of the ABD domain (RV104-ELP2-RV107) morbidity and mortality could not be prevented or delayed, whereas treatment with both RV104-ELP2$_{ABD}$-RV107 and RV104-ELP2$_{ABD}$-RV150 resulted in a marked delay in mortality, of which the delay was most pronounced for RV104-ELP2$_{ABD}$-RV107 (*Figure 5I*). Interestingly, the trispecific RV104-ELP-RV150-ELP$_{ABD}$-RV107 complex protected the mice most efficiently, thereby suggesting that targeting of up to three antigenic sites simultaneously is very effective.

For SBV, the SB10-ELP2$_{ABD}$-SB10 complex prevented mortality in 5 out of 6 mice whereas the SB9-ELP2$_{ABD}$-SB10 complex protected all of the inoculated mice (*Figure 5J*). In both groups, none of the surviving animals showed any clinical signs of disease or significant weight loss. In addition, for all the antibody treated mice viral RNA loads in blood and tissue samples were markedly reduced in comparison to the control animals (*Figure 5—figure supplement 1*).

## Formation and *in vivo* potency of bispecific llama-human chimeric antibodies

To improve the potency of the RVFV VHHs, we designed and constructed a llama-human chimeric bispecific format that encompass effector functions similar as conventional immunoglobulins. Moreover, we aimed to generate HCAbs that are suitable for therapeutic application in humans and with good manufacturability. We started with a human IgG$_1$ CH2 and CH3 backbone, which enables binding to complement factor C1q and Fc receptors of immune cells triggering their activation. The interface between the CH2–CH3 domains also contains the binding site for the FcRn, responsible for the prolonged half-life of IgG, placental passage, and transport of IgG through mucosal surfaces (*Vidarsson et al., 2014*). Several bivalent and/or bispecific hIgG$_1$Fc-VHH fusions were constructed (*Figure 6A,B*) and produced at small scale in HEK 293 T cells (*Laventie et al., 2011*). In line with the ELP results, efficient neutralization at $ND_{50}$ <1 nM was mainly observed with the bispecific constellations (*Figure 6C*) despite binding avidities to Gn$_{ecto}$ that were highly similar (*Figure 6D*). The two most promising bispecific Fc fusions, 150-hIgG$_1$Fc-104 and 107-hIgG$_1$Fc-104 were subsequently produced at medium scale using a Chinese Hamster Ovary (CHO) cell-based transient expression platform (20 L) in disposable rocking bioreactors (*Daramola et al., 2014*). The purified material was subsequently tested in the RVFV BALB/c mouse model. In addition to prophylactic administration, therapeutic efficacy was also evaluated (*Figure 6E*). The results show that 60-100% protection was achieved with prophylactic administration and, remarkably, 60% protection was achieved after therapeutic administration (*Figure 6F,G*). Altogether, these results emphasize that VHH-based biotherapeutics hold great promise for the treatment of bunyavirus infections either prophylactically or therapeutically.

## Discussion

Bunyaviruses pose a clear and present danger to both animal and human health. The WHO recognized the threat for human health by including three bunyaviruses; RVFV, Crimean-Congo hemorrhagic fever virus (CCHFV) and severe fever with thrombocytopenia syndrome virus (SFTSV), on the

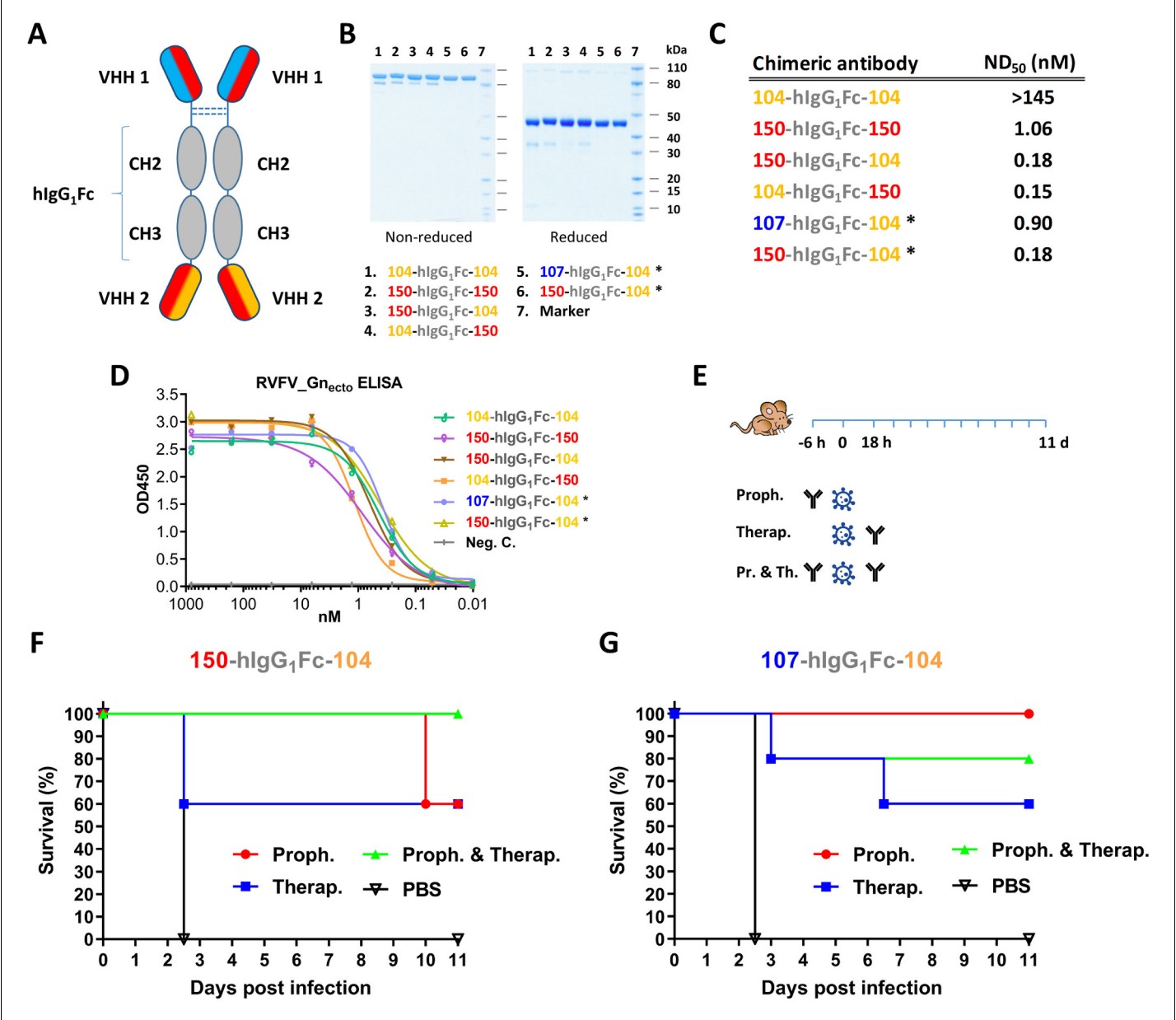

**Figure 6.** Llama-human chimeric antibodies protect mice from lethal RVFV infection. (A) Illustration of the VHH and hIgG$_1$Fc-based llama-human chimeric antibodies. (B) SDS-PAGE of the purified chimeric antibodies under non-reducing and reducing conditions. (C) Neutralizing activity of the purified chimeric antibodies expressed as ND$_{50}$. (D) Indirect RVFV-Gn$_{ecto}$-based ELISA of the chimeric antibodies. (E) Illustration of the mouse experiment setup. (F) Survival curve of RVFV-infected mice treated with 200 µg 150-hIgG$_1$Fc-104. (G) Survival curve of RVFV-infected mice treated with 200 µg 107-hIgG$_1$Fc-104. * These chimeric antibodies were produced at medium scale in CHO cells.

Blueprint list of viruses likely to cause future epidemics and for which no countermeasures are currently available (https://www.who.int/blueprint/priority-diseases/en/). Although for some bunyaviruses veterinary vaccines are available or in development, for humans neither registered vaccines nor effective therapies exist.

The lack of knowledge about bunyavirus glycoprotein structures and immunogenic domains together with rudimentary knowledge on mechanisms of neutralization, have hampered the development of antibody-based bunyavirus therapies. However, novel structural data on the glycoproteins, obtained from combined X-ray and cryo-electron microscopy/tomography of whole virions for some members of the *Bunyavirales* provide now exciting new opportunities (*Wu et al., 2017b*;

*Hellert et al., 2019*; *Halldorsson et al., 2018*; *Allen et al., 2018*). For RVFV and SBV, a limited number of conventional mouse and rabbit mAbs as well as human antibodies have been described with neutralizing activity (*Wu et al., 2017b*; *Hellert et al., 2019*; *Besselaar and Blackburn, 1992*; *Allen et al., 2018*; *Wang et al., 2019*; *Wernike et al., 2015a*). With this study, i) we provide a whole panel of VHHs targeting RVFV or SBV, ii) show neutralizing activity of VHH-based complexes *in vitro* and *in vivo* and, iii) provide a broadly applicable method for the development of VHH-based biotherapeutics.

We specifically harnessed the advantageous characteristics of VHHs in combination with bacterial superglues to develop highly potent virus neutralizing complexes. A major advantage of the presented approach over more conventional approaches is the modularity and subsequent flexibility and directionality in complex construction, making selection of the most potent complexes a relatively easy task. This modularity additionally ensures efficient production, while genetic fusion of VHHs generally results in decreased production yields. Moreover, interference with CDRs is limited due to the flexibility in positioning of the superglue tags in the VHHs, particularly when compared to VHHs linked by genetic fusion, where N-terminal fusions can decrease affinity (*Els Conrath et al., 2001*). Multimeric complexes have not only proven effective in neutralizing viruses, but the presented bacterial superglue and module-based assembly strategy may also facilitate the development of biotherapeutics targeting cancer and toxins. Interestingly, a related approach primarily based on SpyTag-SpyCatcher interactions has recently shown promise to functionally inactivate the human epidermal growth factor receptor 3 (HER3) complex (*Alam et al., 2018*). A potential disadvantage of using bacterial superglues in therapeutics is their intrinsic immunogenicity, leading to immune responses that do not contribute to protection and may even reduce efficacy, particularly with successive treatments. Nevertheless, the bacterial superglue strategy finds its added value, as we showed, as a very efficient intermediate step for identification of potent VHH combinations. After optimal combinations of VHHs are identified, these can be used to develop bispecific constructs with human IgG$_1$ Fc domains.

One of the most profound observations we made in our study was the very strong synergistic effects in neutralization upon targeting of two or three different antigenic sites simultaneously. The synergistic effect of mixtures of antibodies targeting two or more distinct antigenic sites on RVFV virions was already reported more than 25 years ago by *Besselaar and Blackburn (1992)*. They showed that conventional non-neutralizing RVFV monoclonal antibodies efficiently neutralized when they were used in combination (*Besselaar and Blackburn, 1992*). The requirement to target more than one antigenic site for effective bunyavirus neutralization might be related to the broad range of receptor and attachment molecules used for viral attachment and entry in both insect and mammalian cells of which several are of non-proteinaceous nature (*Albornoz et al., 2016*). Remarkably, we did not observe efficient neutralization by simply mixing VHHs targeting two distinct antigenic sites. Efficient neutralization required conjugation of the VHHs to a larger scaffold. Only these larger complexes were found to reduce or even prevent virus infection. Several mechanisms could explain the higher efficacy of multimeric complexes, including more efficient interference with receptor binding or interference with fusion through blocking viral glycoprotein conformational changes. Cross-linking of the glycoproteins may also reduce their overall degrees of freedom thereby preventing the glycoprotein shell to disassemble, which is a crucial step in the fusion process. Finally, binding of the complexes to multiple sites at the virion surface may facilitate complement activation and/or opsonization.

To gain insight into the mechanisms of RVFV neutralization, the VHHs were mapped on the Gn glycoprotein. This showed that all selected RVFV VHHs bind to the head domain of the Gn glycoprotein. In the virion context, Gn forms heterodimers with Gc and assembles into capsomers that are organized into an icosahedral lattice with a T = 12 quasisymmetry (*Halldorsson et al., 2018*; *Sherman et al., 2009*; *Huiskonen et al., 2009*). Gn is involved in receptor binding and Gc, as a class II fusion protein, is responsible for fusion of the viral membrane with the endosome (*Guardado-Calvo et al., 2017*; *Kielian, 2006*). To prevent premature fusion of Gc, the Gc fusion peptide is shielded by Gn subdomain A (I) and the β-ribbon domain (II) (*Halldorsson et al., 2018*). Binding of VHHs that target the β-ribbon domain (blue panel) could potentially interfere with heterodimer dissociation for fusion or with fusion loop insertion. The VHHs targeting domain B (yellow panel) most likely interfere with receptor binding although specific functions have yet to be identified for this

domain. Since the red VHH panel was unable to bind to any of the subdomains we expect these to bind at a domain interface.

The competition observed between the SBV-specific VHHs revealed that the majority (except for SB10) bind to one of three overlapping antigenic sites. X-ray crystallography studies previously showed that mAb 1C11 binds to the apical end of the Gc head domain (*Hellert et al., 2019*). Since SB11 and SB14 partially block antigen recognition by mAb 1C11, although not reciprocally, it appears that the antigenic sites recognized by the majority of our VHHs partially overlap with the antigenic site of mAb 1C11. The apical end of the Gc head domain thus appears to be highly immunogenic in llamas. A fourth, independent antigenic site is defined by SB10, for which an escape mutation at the hinge that connects the head domain to the stalk domain was identified. Intriguingly, the same point mutation was also found in an SBV field sample from a malformed goatling in 2012 (BH 233/12–1, GenBank: KC108871, *Fischer et al., 2013*). The virus in this goatling possibly replicated in the face of sub-neutralizing levels of antibodies that selected for this mutation. The mutation could affect the binding of the antibodies directly or the mutation could alter the conformation of Gc, leading to less efficient neutralization by SB10 or similar antibodies. We postulate that antibodies similar to SB10 are produced by more than just one natural target species, and that these antibodies pose a significant fitness cost for viral growth. As the biological role of the SBV Gc head domain is not yet fully understood, mechanisms for antibody protection remain speculative at this point. It becomes however apparent that binding of different antibodies to the SBV Gc head domain may alter the tertiary or quaternary structure of the spike and influence its structural dynamics: whereas at least two antigenic sites have been mapped to the putative trimerization interface of the head domain, SB10 binding likely restrains the head domain's mobility at its connection to the stalk. Together with inhibition of host cell attachment, these mechanisms may well contribute to virus neutralization.

Following the proof of principle experiments with the ELP-based complexes, selected VHHs were reformatted to bispecific hIgG$_1$Fc-VHH fusion proteins (*Laventie et al., 2011*). In contrast to the ELP-based VHH complexes, the hIgG$_1$Fc bispecific format is potentially capable of eliciting effector function and expected to have a prolonged half-life in plasma due to FcRn binding. Orthologous mouse and human FcγRs share ~60–70% identity, suggesting some incompatibility. However, a recent study shows that hIgG$_1$ binds all mouse FcγRs with affinities that are similar to binding affinities of mouse IgG$_{2a}$ to the mouse receptors (*Dekkers et al., 2017*). Hence, FcγR-mediated effector functions of the hIgG$_1$Fc based bispecifics are expected to be maintained in mice. Production of the two hIgG1Fc-VHH fusion proteins with the CHO cell based transient expression platform (*Daramola et al., 2014*) was efficient and preliminary data suggest that the expression and purification profiles are similar to what is usually seen with conventional bispecific antibodies. Nevertheless, a more comprehensive characterization of the molecules is still required to fully assess the manufacturability of this new bispecific format.

Altogether, our study describes a novel method to rapidly screen and format VHHs into highly potent multispecific complexes, which besides providing ammunition in the battle against (bunya) viruses, is expected to provide novel opportunities for the development of cancer and toxin biotherapeutics.

# Materials and methods

**Key resources table**

| Reagent type (species) or resource | Designation | Source or reference | Identifiers | Additional information |
|---|---|---|---|---|
| Strain, strain background (virus, *Ovis aries*) | RVFV-35/74 | (*Kortekaas et al., 2011*) | RVFV-35/74 | Recombinant virus |
| Strain, strain background (virus, *Homo sapiens*) | RVFV-Clone 13 | (*Muller et al., 1995*) | RVFV-Clone 13 | Natural isolate lacking 69% of the NSs gene |

*Continued on next page*

Continued

| Reagent type (species) or resource | Designation | Source or reference | Identifiers | Additional information |
|---|---|---|---|---|
| Strain, strain background (virus, *Bos taurus*) | SBV NL-F6 | (*Van Der Poel et al., 2014*; *Hulst et al., 2013*) | SBV NL-F6 | Natural isolate |
| Strain, strain background (virus, *Ovis aries*) | SBV BH619 | (*Wernike et al., 2015b*) | SBV BH619 | Natural isolate |
| Strain, strain background (*E. coli*) | TG1 cells | Immunosource | 60502–2 | Electrocompetent cells |
| Strain, strain background (*E. coli*) | BL21 (DE3) | New England Biolabs | C2527H | Competent cells |
| Strain, strain background (*Mus musculus*) | BALB/cAnNCrl mice | Charles River Laboratories | BALB/cAnNCrl | |
| Strain, strain background (*Mus musculus*) | IFNAR-/-mice C57BL/6 | FLI | B6.129S2-Ifnar 1tm1Agt/Mmjax | |
| Strain, strain background (yeast) | *S. cerevisiae* | (*Harmsen and De Haard, 2007a*) | *S. cerevisiae* | |
| Cell line *Chlorocebus aethiops* | Vero E6 | ATCC | CRL-1586 | |
| Cell line *Spodoptera frugiperda* | Sf9-ET cells | ATCC | CRL-3357 | |
| Cell line *Trichoplusia ni* | High Five cells | Thermo Fisher Scientific | B855-02 | |
| Recombinant DNA reagent | pRL144 | (*Harmsen et al., 2005*) | pRL144 | Phage display vector |
| Peptide, recombinant protein | RVFV-Gn$_{ecto}$ | (*de Boer et al., 2010*) | | |
| Peptide, recombinant protein | SBV-Gc$_{head}$ | (*Wernike et al., 2017*) | | |
| Recombinant DNA reagent | Coding regions RVFV-Gn-ecto | This paper | Genscript | *Table 1* |
| Recombinant DNA reagent | Coding region SBV-Gc-head | This paper | Genscript | *Table 1* |
| Recombinant DNA reagent | pRL188 | (*Harmsen et al., 2007b*) | AJ811567 | Yeast expression vector |
| Recombinant DNA reagent | pQE-80L | Qiagen | | Expression plasmid |
| Recombinant DNA reagent | pBAC3 | Merck | 70088 | Baculo transfer plasmid |
| Commercial assay, kit | ELISA Streptactin coated microplates | IBA Lifesciences | 2-1501-001 | |
| Commercial assay, kit | FlashBac ULTRA system | Oxford Expression Technologies | 100300 | |
| Commercial assay, kit | Lightning-Link HRP Conjugation Kit | Innova Biosciences | AB102890 | |
| Other | Gravity Flow Strep-tactin Sepharose column | IBA | 2-1202-001 | |

*Continued*

| Reagent type (species) or resource | Designation | Source or reference | Identifiers | Additional information |
|---|---|---|---|---|
| Other | Amicon Ultra centrifugal filters | Merck Millipore | UFC900324 | |
| Other | RVFV VLPs | (*de Boer et al., 2010*) | | Virus-like particles |
| Other | Ni-NTA resin | Qiagen | 30210 | |
| Other | Protein A agarose Fast Flow 50% | Sigma | P3476 | |
| Other | CHO transient expression system | (*Daramola et al., 2014*) | | Expression system |
| Other | Human albumin | Sigma | A9511 | |
| Other | Mouse albumin | Sigma | A3139 | |
| Other | Bovine albumin | Sigma | A7906 | |
| Other | Bis-Tris NuPAGE Novex Gels | Life Technologies | 4–12% NP0322 12% NP0342 | |
| Other | TMB One Component HRP Microwell Substrate | SurModics | TMBW-1000–01 | |
| Other | HRP-conjugated Strep-Tactin | IBA | 2-1502-001 | 1:5000 |
| Commercial assay, kit | QIAamp Viral RNA kit | Qiagen | 52904 | |
| Commercial assay, kit | RNA Clean and Concentrator −5 kit | Zymo | R1013 | |
| Commercial assay, kit | Phusion Flash High-Fidelity PCR Master Mix | Thermo Fisher Scientific | F548 | |
| Commercial assay, kit | MagAttract Virus mini M48 kit | Qiagen | 955336 | |
| Commercial assay, kit | DNA Clean and Concentrator-5 kit | Zymo | D4014 | |
| Commercial assay, kit | Superscript III First-Strand Synthesis System | Invitrogen | 18080051 | |
| Sequence-based reagent | JR565 | This paper | PCR primers | GACAATTGATG ACACATATAGCTT |
| Sequence-based reagent | JR829 | This paper | PCR primers | ACAGAGCCTCTG AGAAATGTCTG |
| Sequence-based reagent | JR830 | This paper | PCR primers | GATTTGCATACC AGTATTGGTG |
| Antibody | Polyclonal HRP-conjugated goat anti-llama IgG-H+L | Bethyl | A160-100P | IPMA (1:1000), ELISA (1:2000) |

## Viruses, cells and media

Culture media and supplements were obtained from Gibco (Life Technologies, Paisley, United Kingdom) unless indicated otherwise. Virus stocks of RVFV strain Clone 13 (*Muller et al., 1995*) and SBV strain NL-F6 (*Van Der Poel et al., 2014*) were obtained after infections at low multiplicity of infection (0.01) of Vero E6 cells (ATCC CRL-1586, Teddington, United Kingdom). Vero cells were maintained in Eagle's minimal essential medium (EMEM) supplemented with 1% nonessential amino acids (NEAA), 1% antibiotic/antimycotic (a/a) and 5% foetal bovine serum (FBS), at 37°C with 5% $CO_2$. Cells were regularly tested and were mycoplasma free.

## Llama immunization and phage display selection

Two adult llamas (*Lama glama*, #9342, #9343) were intramuscularly (hind leg) immunized with 1 ml of RVFV strain Clone 13 ($10^7$ $TCID_{50}$) and another two llamas (#9340, #9341) were inoculated via the

**Table 1.** Amino acid sequences of used domains and tags.

| Protein | Amino acid sequence |
| --- | --- |
| SBV Gc<sub>head</sub> | INCKNIQSTQLTIEHLSKCMAFYQNKTSSPVVINEIISDASVDEQELIKSLNLNCNVIDRFISESSVIETQV YYEYIKSQLCPLQVHDIFTINSASNIQWKALARSFTLGVCNTNPHKHICRCLESMQMCTSTKTDHARE MSIYYDGHPDRFEHDMKIILNIMRYIVPGLGRVLLDQIKQTKDYQALRHIQGKLSPKSQSNLQLKGFL EFVDFILGANVTIEKTPQTLTTLSLI |
| Twin Strep-tag | GSAWSHPQFEKGGGSGGGSGGGSAWSHPQFEK |
| Yeast signal peptide | MMLLQAFLFLLAGFAAKISA |
| SpyTag | AHIVMVDAYKPTK |
| SpyCatcher | DIPTTENLYFQGAMVDTLSGLSSEQGQSGDMTIEED SATHIKFSKRDEDGKELAGATMELRDSSGKTIS TWISDGQVKDFYLYPGKYTFVETAAPDGYEVATAITFT VNEQGQVTVNGKATKGDAHIDGPQGIWGQLEWKK |
| SnoopTag | KLGDIEFIKVNK |
| SnoopCatcher | SSGLVPRGSHMKPLRGAVFSLQKQHPDYPDIYGAI DQNGTYQNVRTGEDGKLTFKNLSDGKYR LFENSEPAGYKPVQNKPIVAFQIVNGEVRDVTSIVP QDIPATYEFTNGKHYITNEPIPPKGPQGIWGQLDGHGVG |
| ELP (1-2-3) | NL(GVPGVGVPGVGVPGEGVPGVGVPGVGVPGVGVPGVGVPGEGVP GVGVPGVGVPGVGVPGVGVPGEGVPGVGVPGVGVPG)<sub>1-3</sub>GLL |
| HisTag | MKGSSHHHHHH |
| SdyCatcher | SSGLVPRGSHMASMTGGQQMGRGSSGLSGETGQSGNTTIEEDSTTHVK FSKRDANGKELAGAMIELRNLSGQTIQSWISDGTVKVFYLMPGTYQFVE TAAPEGYELAAPITFTIDEKGQIWVDS |
| SdyTag | DPIVMIDNDKPIT |
| RVFV-Gn-A | EDPHLRNRPGKGHNYIDGMTQEDATCKPVTYAGACSSFDVLLEKGK FPLFQSYAHHRTLLEAVHDTIIAKADPPSCDLQSAHGNPCMKEKLVM KTHCPNDYQSAHYLNNDGKMASVKCPPKYELTEDCNFCRQMT GASLKKGSYPLQ |
| RVFV-Gn-ß | QDLFCQSSEDDGSKLKTKMKGVCEVGVQALKKCDGQLSTAHEVVP FAVFKNSKKVYLDKLDLKTEENGSGSGVVQIQVSGVWKKPLCVG YERVVVKRELSA |
| RVFV-Gn-B | NLLPDSFVCFEHKGQYKGTMDSGQTKRELKSFDISQCPKIGG HGSKKCTGDAAFCSAYECTAQYANAYCSHANGSG |

same route with SBV NL-F6 ($10^6$ TCID$_{50}$). Four weeks post immunization the VHH repertoire of the individual animals was amplified by RT-PCR from peripheral blood lymphocytes and inserted into phage display vector pRL144, as earlier described (*Harmsen et al., 2005*). Additionally, a library was taken along from a llama (#3467) that was three times immunized with 20 µg RVFV VLPs (*de Boer et al., 2010*) using a four week interval between immunizations 1 and 2 and a three week interval between immunizations 2 and 3. A week post the second vaccination a blood sample was taken for library preparation. Of note, simultaneously with the RVFV VLP antigen, this llama was also immunized with influenza antigens for a different study (*Harmsen et al., 2013*). Libraries consisting of at least $10^7$ unique clones were generated and phages were rescued (*Harmsen et al., 2007b*). Phage display selections were subsequently performed by consecutive rounds of biopanning using RVFV-Gn<sub>ecto</sub> and SBV-Gc<sub>head</sub> expressed with a N-terminal Twin-Strep-tag as a bait antigen coated to Strep-Tactin coated microplates (IBA Lifesciences, Göttingen, Germany). To assess the enrichment of phages displaying RVFV-Gn<sub>ecto</sub>- or SBV-Gc<sub>head</sub>-specific VHHs, parallel phage ELISA and phage display selection were performed. After panning round two or three, individual clones were picked and grown in suspension for small-scale VHH production.

## Isolation and identification of RVFV and SBV-specific VHHs

After the second or third round of panning and transduction to *E. coli* TG1 cells, individual colonies were picked and the VHH expression was induced with 3 mM IPTG. Soluble VHHs, extracted from the periplasm, were tested for binding to the RVFV-Gn<sub>ecto</sub> and SBV-Gc<sub>head</sub> antigens at 10-fold

dilution as described below (ELISA). Individual VHHs were sequenced and aligned according to the IMGT system as described (*Harmsen et al., 2000*; *Lefranc et al., 2003*).

## Production of RVFV-Gn~ecto~ and SBV-Gc~head~ antigen

The coding regions of the RVFV-Gn$_{ecto}$ (and smaller subdomains) and the SBV-Gc$_{head}$ domain (*Table 1*) were gene synthesized by GenScript (Piscataway, New Jersey), and cloned into the pBAC3 baculo transfer plasmid (Merck) resulting in the following open reading frame (ORF) organization: signal sequence GP 64 – SpyCatcher – 10 GlySer linker – *domain of interest* – enterokinase - Twin Strep-tag. After rescue of the recombinant baculoviruses using the FlashBAC ULTRA system (Oxford Expression Technologies, Oxford, United Kingdom) in Sf9-ET cells (ATCC CRL-3357) according to the manufacturer's instructions, protein productions were initiated in High Five cells (Thermo Fisher Scientific, Landsmeer, The Netherlands) according to the manufacturer's instructions. The proteins of interest were subsequently purified from the culture supernatant using gravity flow Strep-tactin sepharose columns (IBA) according to the manufacturer's instructions. Finally, proteins were buffer exchanged to TBS with 200 mM NaCl using Amicon Ultra centrifugal filters (Merck Millipore, Burlington, Massachusetts).

## VHH production

For medium scale production of VHHs a yeast (*S. cerevisiae*) expression system was used. VHH gene fragments were either transferred from pRL144 or synthesized by GenScript and transferred into the yeast expression vector pRL188 (*Harmsen et al., 2007b*). The SpyTagged, SdyTagged and Snoop-Tagged versions of the VHHs were organized as follows: yeast signal sequence – VHH – 15 GlySer linker – Spy/Sdy/SnoopTag – GAA linker – HisTag (*Table 1*). *S. cerevisiae* cells were induced for VHH expression using galactose and VHHs were purified from the culture supernatant by IMAC using Ni-NTA resin as described (*Harmsen et al., 2007b*). The buffer was exchanged to PBS using 3 kDa Amicon Ultra centrifugal filters (Merck Millipore).

## ELP scaffolds

Scaffold sequences of three different lengths (*Table 1*), based on previous work of Howarth and colleagues (*Zakeri et al., 2012*; *Veggiani et al., 2016*; *Li et al., 2014*), were codon-optimized for expression in *E. coli* and synthesized by GenScript. The gene fragments were cloned into the pQE-80L expression plasmid (ORF organization: HisTag – SnoopCatcher – ELP(1/2/3) – SpyCatcher – GlySer linker- ABD – Entero kinase- Twin Strep-tag) (Qiagen, Hilden, Germany). For the expression of the scaffold that facilitated formation of a trispecific VHH complex the ORF organization was as follows: HisTag – SnoopCatcher – ELP – SpyCatcher – ELP - SdyCatcher – GlySer linker- ABD – Entero kinase – Twin Strep-tag. Plasmids were subsequently introduced into *E. coli* BL21 competent cells by standard procedures. Transformed colonies were grown in Luria-Bertani (LB) broth and induced by IPTG using standard procedures. Cells were subsequently lysed under denaturing conditions, using 6 M guanidine hydrochloride as a denaturant. The scaffolds were purified from the supernatants by IMAC using Ni-NTA resin. Finally, the scaffolds were refolded by buffer exchange to PBS using 30 kDa Amicon Ultra centrifugal filters (Millipore).

## Formation and purification of VHH complexes

Bivalent, bispecific and trispecific complexes were formed upon mixing of SpyTagged, Snoop-Tagged and/or SdyTagged VHHs with ELP scaffolds with C-terminally, N-terminally and/or internal Spy-, Snoop- or SdyCatchers. Briefly, scaffold and VHH solutions were diluted in PBS to concentrations between 5–32 µM. Subsequently, solutions were mixed at a 1:3-1:4 molar ratio (scaffold:VHH) under agitation (300 rpm) in a thermoblock (Eppendorf, Hamburg, Germany) for 3 hr at 20°C. Of note, for the trispecific complex the SdyTagged VHHs were incubated 3 hr at 20°C before the Spy- and SnoopTagged VHHs were added. When required, complexes were separated from uncoupled VHHs by Strep-Tactin based affinity chromatography according the manufacturer's instructions (IBA).

## Construction and small-scale production of hIgG$_1$Fc-VHH chimeric antibodies

Tetravalency and/or bi-specificity was achieved by cloning one VHH in frame with the human IgG$_1$ hinge, CH2 and CH3 exons (*Laventie et al., 2011*). The stop codon was removed to allow translation into the other VHH domain via a flexible synthetic linker (ERKPPVEPPPPP). The secreted antibody is a dimer of ~110 kDa (*Figure 6B*). For a small scale production we stably transfected HEK 293 T cells with the resulting plasmid (pCAG hygro G1 variant) and purified bispecific antibodies from the supernatant using protein A affinity resin as described previously for HCAbs (*Drabek et al., 2016*).

## Production of hIgG$_1$Fc-VHH chimeric antibodies in CHO transient expression system

For medium scale production, the hIgG$_1$Fc-VHH chimeric antibodies were expressed in a CHO transient expression system as previously described (*Daramola et al., 2014*). The sequences for the hIgG$_1$Fc-VHH chimeric antibodies were synthesized by GeneArt, ThermoFisher and cloned into OriP-containing expression vectors (*Gahn and Sugden, 1995*). Twenty litre cell culture volumes in disposable rocking bioreactors were subsequently transiently transfected with the expression vectors. The clarified harvest supernatant was purified using Protein A-based chromatography (*Liu et al., 2010*).

## SDS-PAGE

Samples were denatured using SDS-PAGE sample buffer with or without DTT and heated for 7–10 min at 95˚C. Generally, samples containing 2.5–3 µg of total protein were loaded onto 4–12% or 12% Bis-Tris NuPAGENovex Gels (Life Technologies). Gels were stained with GelCode Blue Stain Reagent (Thermo Scientific, Waltham, Massachusetts) according to the manufacturer's instructions.

## HRP conjugation of VHHs

Purified VHHs were conjugated with HRP using the Lightning-Link HRP Conjugation Kit (Innova Biosciences, Cambridge, United Kingdom).

## ELISAs

Strep-Tactin microplates (Cat. No. 2-1501-001, IBA) were coated with 100 µl/well of RVFV-Gn$_{ecto}$ or SBV-Gc$_{head}$ at 1 µg/mL in ELISA binding buffer (25 mM Tris-HCl, 2 mM EDTA, 140 mM NaCl, pH 7.6) overnight at 4˚C and then washed with PBS supplemented with 0.05% Tween 20 (PBST$_{20}$) using an ELISA washer (6 pulses). Plates were blocked with 300 µl/well of ELISA blocking buffer (2% w/v skimmed milk in PBST$_{20}$) for 1 hr at RT. Plates were subsequently incubated with 100 µl/well of five-fold dilution series in blocking buffer of unconjugated (indirect ELISA) VHHs, VHH complexes or VHH-hIgG$_1$ fusions (starting at 10 µg/ml) for 1 hr at RT and then washed with the ELISA washer. HRP-conjugated goat anti-llama IgG-H+L (A160-100P, Bethyl, Montgomery, Texas) diluted 1:2000 in blocking buffer (100 µl/well) was used as a secondary antibody (1 hr at RT). For the competition ELISA, plates were loaded with 90 µl/well of the competing VHHs at 11.11 µg/ml in blocking buffer, followed by incubation for 1 hr at RT. Subsequently, 10 µl/well of the analyte HRP-conjugated VHHs (concentration differs per VHH and should reach E$_{450}$ 0.5–0.8 in indirect ELISA) in blocking buffer were added, followed by incubation for 1 hr at RT. In all three ELISAs, TMB One Component HRP Microwell Substrate (TMBW-1000–01, SurModics, Minnesota) was added as a substrate. For the species-specific albumin ELISA of the ABD containing ELP scaffold, human (A9511 Sigma), mouse (A3139 Sigma) or bovine albumin (A7906 Sigma) was coated at a 10 µg/ml concentration. Plates were blocked with 300 µl/well of casein blocking buffer (2% casein in PBST$_{20}$) for 1 hr at RT. Plates were subsequently incubated with 100 µl/well ABD containing ELP scaffold in a dilution series. Strep-Tactin-HRP conjugate (2-1502-001, IBA) diluted 1:5000 in casein blocking buffer (100 µl/well) was used as a secondary antibody (1 hr at RT).

## Immunoperoxidase monolayer assay (IPMA)

Cells were fixed with 4% v/v paraformaldehyde for 15 min at RT and then washed with PBST$_{20}$. For cell permeabilization, the fixed cells were incubated with PBS supplemented with 1% v/v Triton X-100 for 5 min at RT. After three washes with PBST$_{20}$, plates were subsequently blocked with IPMA

blocking buffer (5% v/v horse serum in PBS) and incubated for 1 hr at 37°C. Diluted primary antibodies (VHHs) in blocking buffer (100 µL/well) were added. Plates were incubated for 1 hr at 37°C and then washed three times with $PBST_{20}$. Subsequently, a HRP-conjugated goat anti-llama IgG H+L secondary antibody (Bethyl, A160-100P) in blocking buffer (1:1000, 100 µL/well) was added. Plates were incubated for 1 hr at 37°C and then washed three times with $PBST_{20}$. For staining, 100 µL/well of a 0.2 mg/mL amino ethyl carbazole (AEC) solution in 500 mM acetate buffer (pH 5.0), 88 mM $H_2O_2$ was added as a substrate. Plates were incubated for 15–30 min at RT. Titers were calculated as the 50% tissue culture infective dose ($TCID_{50}$)/mL using the Spearman-Kärber method.

## SB10 escape mutant selection

An escape mutant was selected after serial passaging of SBV NL-F6 under selective pressure by SB10. Initially, 1 ml of $10^4$ $TCID_{50}$/ml virus suspension was incubated with 1 ml of SB10 solution at 10 µg/ml for 2 hr at RT. The virus-VHH mixture was then inoculated into T25 flasks seeded with $8 \times 10^5$ Vero E6 cells, and subsequently incubated for 48–96 hr at 37°C with 5% $CO_2$. Virus harvested from the culture supernatants and again subjected to SB10-mediated neutralization was used for sequential rounds of passaging. Suspected mutants from the sixth passage exhibiting cytopathic effects were selected for total RNA isolation using the QIAamp Viral RNA kit (Qiagen). Isolated RNA was purified and concentrated using the RNA Clean and Concentrator-5 kit (Zymo, Irvine, California) according to the instructions of the manufacturer. The SuperScript III First-Strand Synthesis System (Invitrogen) in combination with primer JR565 (GACAATTGATGACACATATAGCTT) was used to reverse transcribe the SBV Gc head domain RNA. An SBV $Gc_{head}$-specific PCR product was amplified using the Phusion Flash High-Fidelity PCR Master Mix (Thermo Scientific) in combination with primer JR829 (ACAGAGCCTCTGAGAAATGTCTG) and JR830 (GATTTGCATACCAGTATTGGTG). PCR products were purified and concentrated using the DNA Clean and Concentrator-5 kit (Zymo). Finally, samples were sent to BaseClear (Leiden, The Netherlands) for Sanger sequencing. SBV NL-F6 without being passaged and SBV NL-F6 passaged without prior incubation with SB10 were used as controls.

## Virus neutralization tests

RVFV neutralization was assessed with the use of a highly sensitive VNT test as described (*Wichgers Schreur et al., 2017*). SBV neutralization was assessed with the following method. In standard 96-well cell culture plates, 50 µl of two-fold dilutions of VHHs (starting at approximately 1000 nM) were incubated with 50 µl of a $10^{4.2}$ $TCID_{50}$/ml SBV suspension for 2 hr at RT, before adding them to $1.5 \times 10^4$ Vero E6 cells/well. VHH-SBV complexes and cells were incubated for 48 hr at 37°C with 5% $CO_2$, and subsequently stained using IPMA. The neutralization capacity was calculated as $ND_{50}$.

## RVFV challenge experiments

Six-week-old female BALB/cAnNCrl mice (Charles River Laboratories) were randomly divided into groups of 5 mice, kept in type III filter-top cages under BSL-3 conditions, and allowed to acclimatize for 6 days. The group size was computed (power 80%) with an estimated mortality (including euthanasia at HEP) in the non-treated control group of 95% and an estimated mortality in the treated group of 15–25% http://clincalc.com/stats/samplesize.aspx. Mice were prophylactically (t = −6 hr) or therapeutically (t = 18 hr) treated with purified VHH complexes (150 µg) or antibodies (150–200 µg) via the intraperitoneal route (in 100 µl PBS). Mice were challenged at t = 0 via intraperitoneal route with $10^3$ $TCID_{50}$ of recombinant RVFV strain 35/74 (*Kortekaas et al., 2011*) in 100 µl medium. Challenged mice were closely monitored and humanely euthanized after reaching a humane endpoint.

## SBV challenge experiments

Eighteen male and female IFNAR-/-mice with a C57BL/6 genetic background (B6.129S2-Ifnar1t-m1Agt/Mmjax) were obtained from the specific pathogen free breeding unit of the Friedrich-Loeffler-Institut. The animals were 4 to 6 weeks old and were randomly divided into 3 groups of 6 animals with equally distributed sexes. The group size calculation followed the same assumptions as for the RVFV challenge experiment. 24 hr prior to challenge infection, the mice were intraperitoneally treated with 200 µg (SB10-ELP2$_{ABD}$-SB10) or 100 µg (SB10-ELP2$_{ABD}$-SB9) of purified VHH

complexes diluted in 100 µl PBS/mouse. Control animals received an equal volume of PBS. The mice were subcutaneously infected with SBV strain BH619 (*Wernike et al., 2015b*) ($10^5$ TCID$_{50}$/mouse, diluted in 100 µl PBS). After challenge infection, the animals were closely monitored and weighed daily for 9 consecutive days. On days 3 and 7 post infection, EDTA blood samples were collected for RT-qPCR analyses. Animals showing severe clinical symptoms were immediately humanely euthanized. The remaining animals were euthanized after 21 days. At necropsy, serum and EDTA blood samples as well as tissue samples from liver and spleen were collected. RNA extraction was performed using a KingFisher 96 Flex (Thermo Scientific) and the MagAttract Virus Mini M48 Kit (Qiagen) according to the instructions of the manufacturer. For RT-qPCR analyses a previously described SBV-S-segment-specific assay was used in combination with the beta actin housekeeping gene as internal control (*Bilk et al., 2012*; *Tauscher et al., 2017*). For quantification of viral RNA, an external SBV standard was included.

## Statistical analyses

When appropriate, data was statistically analysed with GraphPad Prism version 8 software. Specific tests used are described in the figure legends.

## Acknowledgements

RVFV Clone 13 was kindly provided by Dr. Michèle Bouloy (Institut Pasteur, France). Marga van Setten is acknowledged for production of the VHHs in yeast, Amy Clarijs for her assistance with the SBV VHH pannings and Mirriam Tacken, Sophie van Oort and Yanyin Lin for the baculo virus expressions. Members of Early Expression and Supply, Purification Process Sciences and Analytical Sciences teams in AstraZeneca are acknowledged for generation of the bispecific human-llama chimeric antibodies. This study was performed as part of the Zoonotic Anticipation and Preparedness Initiative (ZAPI project; IMI Grant Agreement no. 115760), with the assistance and financial support of the Innovative Medicines Initiative (IMI) and the European Commission, and in-kind contributions from EFPIA partners.

## Additional information

### Competing interests

Olalekan Daramola, Sara Rodriguez Conde, Karen Brennan, Dorota Kozub, Maiken Søndergaard Kristiansen, Kieran K Mistry, Ziyan Deng: is an employee of AstraZeneca. The other authors declare that no competing interests exist.

### Funding

| Funder | Grant reference number | Author |
| --- | --- | --- |
| Innovative Medicines Initiative | ZAPI (grant agreement no. 115760) | Paul J Wichgers Schreur<br>Sandra van de Water<br>Michiel Harmsen<br>Erick Bermúdez-Méndez<br>Dubravka Drabek<br>Frank Grosveld<br>Kerstin Wernike<br>Martin Beer<br>Andrea Aebischer<br>Olalekan Daramola<br>Sara Rodriguez Conde<br>Karen Brennan<br>Dorota Kozub<br>Maiken Søndergaard Kristiansen<br>Kieran K Mistry<br>Ziyan Deng<br>Jan Hellert<br>Pablo Guardado-Calvo<br>Félix A Rey<br>Lucien van Keulen<br>Jeroen Kortekaas |

| Ministerio de Ciencia Tecnología y Telecomunicaciones | Student Fellowship (PEM-066-2015-II) | Erick Bermúdez-Méndez |
| Universidad de Costa Rica | Student Fellowship (OAICE-CAB-05-56-2016) | Erick Bermúdez-Méndez |

The funders had no role in study design, data collection and interpretation, or the decision to submit the work for publication.

### Author contributions

Paul J Wichgers Schreur, Conceptualization, Formal analysis, Supervision, Funding acquisition, Validation, Investigation, Visualization, Methodology, Writing - original draft, Project administration, Writing - review and editing; Sandra van de Water, Jan Hellert, Pablo Guardado-Calvo, Conceptualization, Formal analysis, Investigation, Visualization, Methodology, Writing - original draft, Writing - review and editing; Michiel Harmsen, Conceptualization, Formal analysis, Supervision, Investigation, Visualization, Methodology, Writing - original draft; Erick Bermúdez-Méndez, Formal analysis, Investigation, Visualization, Methodology, Writing - review and editing; Dubravka Drabek, Conceptualization, Formal analysis, Funding acquisition, Investigation, Methodology, Writing - original draft; Frank Grosveld, Conceptualization, Formal analysis, Funding acquisition, Methodology, Writing - original draft, Writing - review and editing; Kerstin Wernike, Formal analysis, Validation, Investigation, Methodology, Writing - original draft, Writing - review and editing; Martin Beer, Félix A Rey, Conceptualization, Supervision, Funding acquisition, Methodology, Writing - original draft, Writing - review and editing; Andrea Aebischer, Conceptualization, Formal analysis, Validation, Investigation, Visualization, Methodology, Writing - original draft, Writing - review and editing; Olalekan Daramola, Conceptualization, Formal analysis, Supervision, Funding acquisition, Methodology, Writing - original draft, Writing - review and editing; Sara Rodriguez Conde, Conceptualization, Formal analysis, Supervision, Investigation, Methodology; Karen Brennan, Dorota Kozub, Maiken Søndergaard Kristiansen, Kieran K Mistry, Ziyan Deng, Formal analysis, Investigation, Methodology; Lucien van Keulen, Conceptualization, Formal analysis, Investigation, Methodology, Writing - original draft, Writing - review and editing; Jeroen Kortekaas, Conceptualization, Supervision, Funding acquisition, Validation, Methodology, Writing - original draft, Project administration, Writing - review and editing

### Author ORCIDs

Paul J Wichgers Schreur https://orcid.org/0000-0001-9790-2438
Erick Bermúdez-Méndez https://orcid.org/0000-0001-9241-7898
Kerstin Wernike https://orcid.org/0000-0001-8071-0827
Pablo Guardado-Calvo http://orcid.org/0000-0001-7292-5270
Félix A Rey http://orcid.org/0000-0002-9953-7988
Jeroen Kortekaas https://orcid.org/0000-0002-0329-0176

### Ethics

Animal experimentation: All animal experiments were conducted in accordance with European regulations (EU directive 2010/63/EU) and were in agreement with either the Dutch (the llama immunization experiment and the RVFV challenge experiments) or German Law on Animal Experiments (SBV infection experiment). Permissions were granted by the Dutch Central Authority for Scientific Procedures on Animals (AVD401002016725) and the German Centre for the Protection of Laboratory Animals (LALLF Nr. 7221.3-1-067/17). Specific procedures were approved by the Animal Ethics Committees of Wageningen Research (WR) and the federal state Mecklenburg-Western Pomerania for the experiments conducted in the Netherlands and Germany, respectively.

### Decision letter and Author response

Decision letter https://doi.org/10.7554/eLife.52716.sa1
Author response https://doi.org/10.7554/eLife.52716.sa2

## Additional files

### Supplementary files
• Transparent reporting form

### Data availability
All data generated during this study are included in the manuscript and supporting files.

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
