## [Decision Letter]

Thank you for submitting your article "Multimeric single-domain antibody complexes protect against bunyavirus infections" for consideration by *eLife*. Your article has been reviewed by two peer reviewers, and the evaluation has been overseen by Karla Kirkegaard as the Senior Editor. The reviewers have opted to remain anonymous.

The reviewers have shared their reviews with one another and the Editor has drafted this decision to help you prepare a revised submission.

Summary:

In this manuscript, the authors presented a strategy based on Llama-derived single-domain antibodies (VHHs) for the development of effective bunyavirus therapeutics. Rift Valley fever virus (RVFV) and Schmallenberg virus (SBV) were used as test cases. First, the authors isolated VHHs from a phage display library constructed from B cells of llamas immunized with RVFV vaccine and SBV VLP. While all animals showed virus-specific neutralizing antibody (NAb) titers, panning-derived VHHs showed low and moderate neutralization activities when tested individually and in a mixed form, respectively. Then, the authors utilized the SpyCatcher/SpyTag technology to create multimeric VHHs to test the synergistic effect in neutralization. Indeed, several multimeric VHHs showed greater potency than individual VHHs. Next, the authors created human IgG1Fc-VHH constructs for the best performers in the synergy test using these tethered or "glued" VHHs. CHO-expressed bispecific VHHs showed protection in a mouse challenge model of bunyavirus infection. Thus, this study presents

While none of the individual technical components in this strategy, e.g. llama VHH phage display and library screening, SpyCatcher/SpyTag for antibody coupling, and generation of bispecific antibodies using VHH and human Fc, is novel, their combination provides an elegant working strategy for developing VHH-based antibody therapeutics for an important viral pathogen. However, incorporation of these technical components into a working pipeline may justify some level of novelty.

Essential revisions:

1) Please carefully review the claims and interpretations to ensure that they are justified and that alternative hypotheses are considered. For example, it is stated that: "both the increased affinity due to multimerization as well as the increase in overall size of the VHH complex, resulting in additional steric hindrance, explain why the complexes outcompeted the mixtures of VHHs" suggests that steric hindrance by SpyCatcher/Tag or human Fc is the mechanism of neutralization. Could there not be others?

2) The SpyCatcher/tag and SnoopCatcher/Tag systems (bacterial SuperGlue) present powerful tools for protein engineering that are well-described here. There has been concern that these bacterial proteins would elicit a strong antibody response in humans. Here, the bacterial superglue to combine VHHs in an intermediate step before testing them in a bispecific form with a human Fc. It would be helpful to the community if you could discuss the design of this step, as opposed to the direct testing of VHH pairs in the hIgG1Fc form.

3) The Figure legends, in general, are not sufficient. Readers will have to review Materials and methods to understand each, but still cannot understand completely without significant rewriting.

4) The reviewers took special care to make detailed comments to improve the clarity of the presentation and the accuracy of the interpretations. Many of these are below. In general, please edit carefully to link each figure to the text describing it – this is, of course, especially important in a paper with several groups and different kinds of approaches. One of the strengths of *eLife* is the opportunity to explain data carefully.

a) Figure 1F, G should be combined into Figure 2, because the interpretation of Figure 1F and 1G are made as a part of Figure 2

b) Subsection “Mapping of the SBV VHHs on the SBV-Gc_head_domain”: Authors described that SB11 and SB14 target the same antigenic site as 1C11. However, the blocking with 1C11 still did not block the binding of SB11 or SB14 (Figure 2B), indicating that those VHH bind to antigenic sites different from that for 1C11.

c) Subsection “Mapping of the SBV VHHs on the SBV-Gc_head_ domain” and Discussion paragraph six: Escape mutant for SB10 encoded the mutation of Y541C. It is not determined whether the mutation affects the binding of SB10, or the mutation alters the conformation of Gc, which leads to less efficient neutralization via SB10. Thus, the result does not provide conclusive evidence of binding region for SB10 and this conclusion should be softened.

d) Figure 2E: Although figure legend or Materials and methods do not sufficiently describe the detail, the SB10 neutralizing antibody titer was lower with wt SBV than SBV-Y541C. This does not seem consistent with the description that SBV-Y541C is an escape mutant of SB10. Please explain.

e) Figure 5C: The methods of albumin ELISA is not described.

f) Figure 5—figure supplement 1: Statistical analysis should be provided for the comparison of viral RNA copies in Figure 5—figure supplement 1A and C.

g) "For SBV, the majority of our VHHs binds to one of three overlapping antigenic sites on the…." Authors showed that blocking with two out of six VHH (SB11 and SB14) could partially block the binding of 1C11, but not vice versa. The description is thus not well supported by given results. Please rewrite for clarity and accuracy.

---

## [Author Response]

Essential revisions:1) Please carefully review the claims and interpretations to ensure that they are justified and that alternative hypotheses are considered. For example, it is stated that: "both the increased affinity due to multimerization as well as the increase in overall size of the VHH complex, resulting in additional steric hindrance, explain why the complexes outcompeted the mixtures of VHHs" suggests that steric hindrance by SpyCatcher/Tag or human Fc is the mechanism of neutralization. Could there not be others?

We acknowledge the point of the reviewer that besides interfering with receptor binding by steric hindrance other mechanisms could also explain why efficient neutralization is predominantly observed following multimerization of VHHs. One potential mechanism could involve a more efficient interference with glycoprotein conformational changes by VHH complexes compared to mixtures of VHHs. Bivalent/bispecific VHHs can potentially cross-link the glycoprotein shell at the virion surface, preventing pH-mediated glycoprotein conformational changes required for fusion of the viral and endosomal membranes. Additionally, at least in the in vivo situation, neutralization might be the indirect result of phagocytosis of opsonized virions either with or without a role for complement. In the revised version of the paper we elaborated on the potential mechanisms of neutralization in the Discussion section.

2) The SpyCatcher/tag and SnoopCatcher/Tag systems (bacterial SuperGlue) present powerful tools for protein engineering that are well-described here. There has been concern that these bacterial proteins would elicit a strong antibody response in humans. Here, the bacterial superglue to combine VHHs in an intermediate step before testing them in a bispecific form with a human Fc. It would be helpful to the community if you could discuss the design of this step, as opposed to the direct testing of VHH pairs in the hIgG1Fc form.

We agree that potential anti-bacterial superglue immune responses are unfavourable and we indeed used the bacterial superglues primarily as an intermediate step before designing the bispecific human Fc-based molecules, thereby preventing the need to clone and express a large number of hIgG1Fc molecules. We have more clearly explained the rationale of our approach in the Discussion section.

3) The Figure legends, in general, are not sufficient. Readers will have to review Materials and methods to understand each, but still cannot understand completely without significant rewriting.

We checked all figure legends and added information to facilitate interpretation of the figures without the need to review the Material and methods section.

4) The reviewers took special care to make detailed comments to improve the clarity of the presentation and the accuracy of the interpretations. Many of these are below. In general, please edit carefully to link each figure to the text describing it – this is, of course, especially important in a paper with several groups and different kinds of approaches. One of the strengths of eLife is the opportunity to explain data carefully.

This point is fully acknowledged and we carefully checked and improved when necessary the link between each figure and the corresponding text.

a) Figure 1F, G should be combined into Figure 2, because the interpretation of Figure 1F and 1G are made as a part of Figure 2

Although we understand the point being raised, combining Figure 1F, G with Figure 2 will result in a very large figure with reduced overall readability and interpretation. Moreover, Figure 1F and G are also closely linked to Figure 1C and D. To improve the overall interpretation of both figures and the link between them we added at the end of Figure legend 1 the following sentence: The colour coding of the individual VHHs is based on the outcome of the competition ELISA result as presented in Figure 2A and 2B.

b) Subsection “Mapping of the SBV VHHs on the SBV-Gc_head_ domain”: Authors described that SB11 and SB14 target the same antigenic site as 1C11. However, the blocking with 1C11 still did not block the binding of SB11 or SB14 (Figure 2B), indicating that those VHH bind to antigenic sites different from that for 1C11.

We agree with the reviewer that the blocking pattern is not fully conclusive with regard to binding sites and that binding sites may only partly overlap or do not overlap at all. The non-reciprocal pattern could be explained by the different size of the antibody molecules and steric hindrance. VHHs may be able to “squeeze” between paratopes of 1C11 whereas the VHHs block efficiently binding sites for 1C11 (since its variable domains would need more space to bind), Secondly, the non-reciprocal pattern could also be the result of differences in avidity of the VHHs versus 1C11. Another, more complex explanation for the non-reciprocal blocking might be that SB11 and SB14 specifically bind to the trimer interface, thereby explaining why 1C11, which is known to bind upon trimer dissociation [11], is not able to bind in this situation. Blocking with 1C11, will subsequently prevent oligomerization of the spike but this will not prevent binding of SB11 and SB14 as part of the epitope is still accessible.

c) Subsection “Mapping of the SBV VHHs on the SBV-Gc_head_ domain” and Discussion paragraph six: Escape mutant for SB10 encoded the mutation of Y541C. It is not determined whether the mutation affects the binding of SB10, or the mutation alters the conformation of Gc, which leads to less efficient neutralization via SB10. Thus, the result does not provide conclusive evidence of binding region for SB10 and this conclusion should be softened.

We agree with the reviewer that the mutation could either directly affect the binding of SB10, or alternatively could indirectly influence neutralization by changing the overall Gc conformation. We added this alternative mechanism to the Discussion section and additionally added a question mark to the beige site in Figure 2G to highlight that the mapping is not fully conclusive.

d) Figure 2E: Although figure legend or Materials and methods do not sufficiently describe the detail, the SB10 neutralizing antibody titer was lower with wt SBV than SBV-Y541C. This does not seem consistent with the description that SBV-Y541C is an escape mutant of SB10. Please explain.

After checking Figure 2E we confirmed that the data was presented correctly. In the presence of low nM concentration of SB10 (20 nM), WT SBV is neutralized efficiently, whereas no neutralization was observed with the SBV-Y541C mutant in the presence of high concentrations of SB-10 (512 nM). Nevertheless, to improve the interpretation of the figure, without the need to carefully read the Y-axis title we reversed the axis numbers in line with the neutralization data presented in Figure 3.

e) Figure 5C: The methods of albumin ELISA is not described.

In the revised version of the manuscript additional details about the albumin ELISA are provided in the figure legend and the Material and methods section.

f) Figure 5—figure supplement 1: Statistical analysis should be provided for the comparison of viral RNA copies in Figure 5—figure supplement 1A and C.

In the revised version of the manuscript statistical analysis of Figure 5—figure supplement 1 is provided. Since animals were euthanized at different time points post infection, after reaching humane endpoints, a comparison between treated and non-treated animals could not be made in panel C.

g) "For SBV, the majority of our VHHs binds to one of three overlapping antigenic sites on the…." Authors showed that blocking with two out of six VHH (SB11 and SB14) could partially block the binding of 1C11, but not vice versa. The description is thus not well supported by given results. Please rewrite for clarity and accuracy.

We reformulated the statement and description to more accurately reflect the observations and validation of conclusions in the revised version of the manuscript.